# Interleukin-1β Induced Matrix Metalloproteinase Expression in Human Periodontal Ligament-Derived Mesenchymal Stromal Cells under In Vitro Simulated Static Orthodontic Forces

**DOI:** 10.3390/ijms22031027

**Published:** 2021-01-20

**Authors:** Christian Behm, Michael Nemec, Alice Blufstein, Maria Schubert, Xiaohui Rausch-Fan, Oleh Andrukhov, Erwin Jonke

**Affiliations:** 1Division of Orthodontics, University Clinic of Dentistry, Medical University of Vienna, 1090 Vienna, Austria; christian.behm@meduniwien.ac.at (C.B.); michael.nemec@meduniwien.ac.at (M.N.); erwin.jonke@meduniwien.ac.at (E.J.); 2Competence Center for Periodontal Research, University Clinic of Dentistry, Medical University of Vienna, 1090 Vienna, Austria; alice.blufstein@meduniwien.ac.at (A.B.); mimi.schubert@hotmail.com (M.S.); 3Division of Conservative Dentistry and Periodontology, University Clinic of Dentistry, Medical University of Vienna, 1090 Vienna, Austria; xiaohui.rausch-fan@meduniwien.ac.at

**Keywords:** orthodontic tooth movement, static orthodontic forces, human periodontal ligament derived mesenchymal stromal cells, matrix metalloproteinases, tissue inhibitor of matrix metalloproteinases, interleukin-1β

## Abstract

The periodontal ligament (PDL) responds to applied orthodontic forces by extracellular matrix (ECM) remodeling, in which human periodontal ligament-derived mesenchymal stromal cells (hPDL-MSCs) are largely involved by producing matrix metalloproteinases (MMPs) and their local inhibitors (TIMPs). Apart from orthodontic forces, the synthesis of MMPs and TIMPs is influenced by the aseptic inflammation occurring during orthodontic treatment. Interleukin (IL)-1β is one of the most abundant inflammatory mediators in this process and crucially affects the expression of MMPs and TIMPs in the presence of cyclic low-magnitude orthodontic tensile forces. In this study we aimed to investigate, for the first time, how IL-1β induced expression of MMPs, TIMPs and how IL-1β in hPDL-MSCs was changed after applying in vitro low-magnitude orthodontic tensile strains in a static application mode. Hence, primary hPDL-MSCs were stimulated with IL-1β in combination with static tensile strains (STS) with 6% elongation. After 6- and 24 h, MMP-1, MMP-2, TIMP-1 and IL-1β expression levels were measured. STS alone had no influence on the basal expression of investigated target genes, whereas IL-1β caused increased expression of these genes. In combination, they increased the gene and protein expression of MMP-1 and the gene expression of *MMP-2* after 24 h. After 6 h, STS reduced IL-1β-induced MMP-1 synthesis and *MMP-2* gene expression. IL-1β-induced *TIMP-1* gene expression was decreased by STS after 6- and 24-h. At both time points, the IL-1β-induced gene expression of *IL-1β* was increased. Additionally, this study showed that fetal bovine serum (FBS) caused an overall suppression of IL-1β-induced expression of MMP-1, MMP-2 and TIMP-1. Further, it caused lower or opposite effects of STS on IL-1β-induced expression. These observations suggest that low-magnitude orthodontic tensile strains may favor a more inflammatory and destructive response of hPDL-MSCs when using a static application form and that this response is highly influenced by the presence of FBS in vitro.

## 1. Introduction

Orthodontic tooth movement (OTM) plays a pivotal role in the treatment of malocclusions and is ultimately induced by applying mechanical strain [1,2]. According to the classic “pressure-tension” hypothesis, mechanical load induces a shift of the teeth, causing a compression zone in the direction of force application and a tension zone at the opposite site [3,4]. This creates a local aseptic inflammation, resulting in bone resorption at the compression site and bone formation at the tension site [2]. The fundamental structure for sensing and translating orthodontic forces from teeth to the alveolar bone is the periodontal ligament (PDL) [5], a highly organized connective tissue [6]. Besides bone resorption and formation, the PDLs’ extracellular matrix (ECM) undergoes continuous remodeling during OTM. This ongoing remodeling is mainly orchestrated by the cellular component of the PDL [5,7,8,9,10], which consists of a heterogeneous fibroblast-like population of mesenchymal stromal cells (hPDL-MSCs). These non-hematopoietic, multipotent progenitor cells [11,12,13] were first isolated from the PDL by Seo et al. in 2004 [14,15]. They are characterized by their self-renewal and tri-lineage differentiation potential, as well as by a specific surface marker expression profile and, consequently, fulfil the minimal criteria for MSCs [11,12,13]. Residing in the perivascular area of the PDL [16,17], hPDL-MSCs are highly involved in PDL homeostasis and directly respond to applied orthodontic forces [18]. It is well established that hPDL-MSCs produce and secrete multiple growth factors, prostaglandins and ECM remodeling enzymes during OTM [5,7,8,9,10].

Various proteolytic enzymes, such as matrix metalloproteinases (MMPs) are involved in remodeling ECM proteins within the PDL [5,8,19]. The MMPs belong to a zinc-dependent proteolytic enzyme family which is divided into six different groups depending on their substrate specificity [20,21,22]. Numerous in vitro studies already investigated the effect of mechanical forces on the expression of various MMPs in the human PDL, mainly focusing on the collagenase MMP-1 and the gelatinase MMP-2 [5,23,24,25,26,27]. These studies mainly showed increased MMP-1 expression in PDL fibroblasts, independently of force type and its static or cyclic application [5,8,24,28,29]. MMP-2 expression, however, is increased in PDL cells after applying tensile forces [5,24,30], whereas compression forces cause no effect or a decrease in its expression [31,32]. Several in vivo studies detected increased MMP-1 and MMP-2 levels in the gingival crevicular fluid (GCF) at the tension as well as compression side shortly after applying orthodontic forces [19,33,34,35]. Additionally, studies demonstrated changes in the constitutive expression of various local tissue inhibitors of metalloproteinases (TIMPs) in human PDL cells after applying mechanical forces. TIMP-1 expression mainly increased in human PDL cells when applying tensile forces [23,30], whereas compressive strain seems to have a declining effect on TIMP-1 expression [36,37]. In vivo studies mainly showed an overall increase in TIMP-1 levels in the GCF during orthodontic treatment [33,34,38].

Interleukin (IL)-1β is an important pro-inflammatory cytokine having several roles in the human periodontium. During orthodontic treatment-induced aseptic inflammation, IL-1β is one of the most abundant cytokines in both the tension as well as the compression zone [38,39]. It directly contributes to extensive alveolar bone remodeling during OTM [40]. Additionally, IL-1β directly influences the continuing turnover of the PDLs’ ECM by regulating the expression of MMPs and TIMPs [41,42,43]. Together with other inflammatory cytokines, such as IL-6 [44], IL-1β is also essential in the bacteria-dependent chronic inflammation of the periodontium [45]. Hence, it is crucial to have a closer look on the combined effects of orthodontic forces and IL-1β on the MMPs and TIMPs expression in PDL cells. Two studies already demonstrated a significant effect of tensile strain on the production of IL-1β-triggered production of MMPs and TIMPs in PDL cells, using cyclic strain application [43,46]. However, no data exist showing the influence of static tensile strain (STS) on IL-1β-dependent introduction of MMPs and TIMPs in PDL cells. The discrimination between these two force application forms is essential since they imitate different orthodontic forces applied to patients by different orthodontic appliances [25,47,48,49].

Therefore, the main aim of the first part of this in vitro study was to investigate the influence of STS on IL-1β-induced expression of MMPs and TIMPs in hPDL-MSCs. In particular, we evaluated the IL-1β-triggered production of MMP-1, MMP-2, TIMP-1 and IL-1β itself in hPDL-MSCs after applying equibiaxial STS with 6% elongation for 6 and 24 h. Secondly, we directly compared the influence of STS on the IL-1β-induced expression of these genes in the absence and presence of fetal bovine serum (FBS). It is essential to evaluate to what extent FBS affects in vitro simulated orthodontic forces on hPDL-MSCs, since it is known that FBS influences these cells during in vitro cultivation [50].

## 2. Results

### 2.1. Immunostaining of Certain Mesenchymal and Hematopoietic Surface Marker in hPDL-MSCs

Appendix A shows the percentage of hPDL-MSCs which expressed certain mesenchymal and hematopoietic surface markers. MSCs surface markers CD29, CD73, CD90 and CD105 were expressed in more than 95% of analysed hPDL-MSCs. The percentage of CD146-positive hPDL-MSCs was about 63.63%. Hematopoietic surface markers CD31, CD34 and CD45 were observed in less than 2.5% of analysed hPDL-MSCs.

### 2.2. Cell Viability

Figure 1 shows representative pictures of live/dead stained and IL-1β treated hPDL-MSCs which were exposed to STS for 24 h. The appearance of green fluorescent living hPDL-MSCs was comparable through all used stimulation conditions. In the absence of STS, IL-1β caused an increased number of red fluorescent dead hPDL-MSCs from 1 ng/mL to 5 ng/mL, independently from FBS concentration. In the presence of STS, a lower number of dead hPDL-MSCs is observed, even in the presence of different IL-1β concentrations. In the presence of 2% FBS, STS caused a slightly higher number of dead hPDL-MSCs in the absence and the presence of 1 ng/mL IL-1β compared to hPDL-MSCs incubated without FBS. Nevertheless, the majority of hPDL-MSCs were viable and no detrimental effect of experimental conditions on cell viability was observed.

### 2.3. MMP-1 Expression

Figure 2 shows the MMP-1 gene and protein expression in hPDL-MSCs in the absence of FBS, 6 and 24 h after applying STS in the absence or presence of different IL-1β concentrations. STS alone did not affect *MMP-1* gene expression levels after 6- and 24-h incubation (Figure 2A,B). On protein level, MMP-1 was also not affected by STS in the absence of IL-1β (Figure 2C,D). After 6- and 24-h incubation, 1 and 5 ng/mL IL-1β significantly increased *MMP-1* gene expression in the absence of STS (Figure 2A,B). In conditioned media, 5 ng/mL IL-1β caused a non-significant increase in MMP-1 protein levels at both time points (Figure 2C,D).

Applying STS caused no effect on IL-1β induced *MMP-1* gene expression after 6 h. After 24 h, however, applying STS significantly increased IL-1β induced MMP-1 gene expression levels (Figure 2A,B). On protein level, STS non-significantly (*p*-value 0.171) counteracted IL-1β-induced protein production after 6 h incubation. After 24 h incubation, IL-1β-induced MMP-1 protein production was further increased by STS, however without any significance (*p*-value 0.248, Figure 2C,D).

Figure 3 shows the MMP-1 gene and protein expression in hPDL-MSCs in the presence of 2% FBS, 6 and 24 h after applying STS in the absence and presence of different IL-1β concentrations. Applying STS in the absence of IL-1β had no significant effect on MMP-1 gene and protein expression levels (Figure 3A–D). In the absence of STS, both IL-1β concentrations significantly increased *MMP-1* gene expression after 6- and 24 h incubation (Figure 3A,B). MMP-1 protein levels were also enhanced by IL-1β in the absence of STS, showing a significant increase after 6 h incubation (Figure 3C,D).

Applying STS to IL-1β-treated hPDL-MSCs significantly increased *MMP-1* gene expression levels after 6 h incubation. 24 h treatment with STS also strengthened IL-1β induced *MMP-1* gene expression, only showing a significant increase in the presence of 5 ng/mL IL-1β (Figure 3A,B). On protein level, STS significantly inhibited IL-1β induced MMP-1 after 6 h stimulation. Applying STS for 24 h causes a further increase in IL-1β triggered MMP-1 protein expression; however, without any (*p*-value 0.198) significance (Figure 3C,D).

### 2.4. MMP-2 Expression

Figure 4 shows the *MMP-2* gene expression in hPDL-MSCs in the absence of FBS, 6 and 24 h after applying STS in the presence of different IL-1β concentrations. Applying STS in the absence of IL-1β had no significant effect on *MMP-2* gene expression levels. After both 6 and 24-h incubation, *MMP-2* gene expression in hPDL-MSCs was significantly enhanced by IL-1β, leading to an overall higher *MMP-2* gene expression level after 24 h incubation (Figure 4A,B). On protein level, MMP-2 was not detectable.

Applying STS for 6 h caused a decrease in IL-1β induced *MMP-2* gene expression, showing a significant decline in the presence of 5 ng/mL IL-1β (Figure 4A). After 24 h, STS strengthened IL-1β induced *MMP-2* gene expression in a concentration-dependent manner, showing a significant rise in the presence of 1 ng/mL IL-1β (Figure 4B).

Figure 5 shows the *MMP-2* gene expression in hPDL-MSCs in the presence of 2% FBS, 6 and 24 h after applying STS in the presence of different IL-1β concentrations. STS alone had no effect on *MMP-2* gene expression (Figure 5A,B). After 6 h incubation*, MMP-2* gene expression was suppressed by both IL-1β concentrations, showing a significant suppression in the presence of 1 ng/mL IL-1β (Figure 5A). No effect of IL-1β on *MMP-2* gene expression was observed after 24 h incubation (Figure 5B). On protein level, IL-1β-induced MMP-2 production was not detectable.

STS counteracted the suppressive effect of IL-1β on *MMP-2* gene expression, causing a significant increase after 6 h incubation (Figure 5A). After 24 h incubation, STS caused no effects on IL-1β-induced *MMP-2* gene expression (Figure 5B).

### 2.5. TIMP-1 Expression

Figure 6 shows the *TIMP-1* gene expression in hPDL-MSCs in the absence of FBS, 6 and 24 h after applying STS in the absence or presence of different IL-1β concentrations. In the absence of IL-1β, STS did not affect *TIMP-1* gene expression after both 6- and 24-h incubation (Figure 6A,B). IL-1β alone enhanced *TIMP-1* gene expression after 6- and 24-h incubation, causing a significant increase in the presence of 5 ng/mL IL-1β at both time points and a concentration-dependent rise 24 h after starting the treatment (Figure 6A,B). On protein level, IL-1β-induced TIMP-1 production was not detectable.

In the presence of IL-1β, applying STS significantly decreased *TIMP-1* gene expression after 6 h incubation. A significant suppression was also observed in the presence of 5 ng/mL IL-1β after 24 h incubation (Figure 6A,B).

Figure 7 shows the *TIMP-1* gene expression in hPDL-MSCs in the presence of 2% FBS, 6 and 24 h after applying STS in the absence or presence of different IL-1β concentrations. In the absence of IL-1β, STS had no significant effect on *TIMP-1* gene expression in hPDL-MSCs at both time points (Figure 7A,B). Both IL-1β concentrations showed no influence on *TIMP-1* gene expression after 6 h incubation (Figure 7A). After 24 h incubation, IL-1β alone enhanced *TIMP-1* gene expression, causing a significant increase with 1 ng/mL IL-1β (Figure 7B). On protein level, IL-1β-induced TIMP-1 production was not detectable.

Applying STS in the presence of IL-1β caused a significant increase in *TIMP-1* gene expression in the presence of both IL-1β concentrations after 6 h incubation (Figure 7A). 24 h after applying STS to IL-1β treated hPDL-MSCs, *TIMP-1* gene expression was significantly enhanced at 5 ng/mL IL-1β (Figure 7B).

### 2.6. IL-1β Expression

Figure 8 shows the *IL-1β* gene expression in hPDL-MSCs in the absence of FBS, 6 and 24 h after applying STS in the absence and presence of different IL-1β concentrations. Applying STS alone caused no detectable *IL-1β* gene expression in hPDL-MSCs, neither after 6- nor after 24-h incubation. After 6- and 24-h stimulation, IL-1β alone caused a concentration-dependent increase in *IL-1β* gene expression (Figure 8A,B). Changes in IL-1β induced IL-1β protein expression were not possible to detect due to the high IL-1β levels used for cell stimulation.

6 h after applying STS in the presence of IL-1β, a strengthening of *IL-1β* gene expression in a concentration-dependent manner was observed, causing a significant higher *IL-1β* gene expression at 5 ng/mL IL-1β compared to 1 ng/mL (Figure 8A). After 24 h incubation, STS caused a significant increase in IL-1β induced *IL-1β* gene expression, independently from the present IL-1β concentrations (Figure 8B).

Figure 9 shows the *IL-1β* gene expression in hPDL-MSCs in the presence of 2% FBS, 6- and 24-h after applying STS in the absence or the presence of different IL-1β concentrations. STS alone caused no detectable *IL-1β* gene expression in hPDL-MSCs, neither after 6- nor after 24-h incubation (Figure 9A,B). After 6 h stimulation, IL-1β caused a concentration-dependent increase in *IL-1β* gene expression (Figure 9A). 24 h incubation with IL-1β alone led to an overall lower *IL-1β* gene expression levels, showing the highest level at 1 ng/mL IL-1β (Figure 9B).

Applying STS in the presence of 5 ng/mL IL-1β caused an increase in *IL-1β* gene expression after 6 h incubation. A significantly higher *IL-1β* gene expression was observed at 5 ng/mL compared to 1 ng/mL IL-1β (Figure 9A). After 24 h incubation, STS increased IL-1β induced *IL-1β* gene expression, showing a significant increase at 5 ng/mL (Figure 9B).

## 3. Discussion

One of the major goals in orthodontic research is to reduce the treatment time by accelerating OTM. Nevertheless, the OTM velocity is limited by biological processes and numerous possibilities are currently under investigation to speed up OTM via affecting these biological pathways [2]. A huge number of in vitro and in vivo studies already demonstrate the involvement of hPDL-MSCs in PDL ECM and bone remodeling during OTM (reviewed in [18,51,52,53]). Within this big data pool, multiple studies investigate the influence of orthodontic forces on MMPs and TIMPs within the PDL [5,23,24,25,26,27,30,36,37]. Several of these studies already evaluated the effect of orthodontic forces on MMPs and TIMPs under inflammatory conditions within the PDL [36,43,46]. All of these studies used a cyclic tensile strain application in vitro, mimicking cyclic orthodontic forces. Such cyclic forces are generated by multibracket appliance in combination with occlusal forces [47]. However, cyclic tensile strain does not reflect orthodontic forces caused by nickel-titanium coil springs [25,48,49]. In this case, static tensile strain is a more appropriate model and therefore, we used it in our in vitro study. The discrimination between these two application forms seems to be essential in vitro, since several studies demonstrated significantly different effects on ECM remodeling enzymes in the human PDL [29,54]. In this study, we demonstrated for the first time that static forces during OTM may favor a more inflammatory and destructive local environment at the tension zone within the PDL when applying low-magnitude forces.

It is well established that cyclic tensile strain executes its function on hPDL-MSCs under inflammatory conditions in a dose-dependent manner [43,55]. High magnitudes increase the expression of pro-inflammatory cytokines, whereas low magnitudes (3–6%) act in an anti-inflammatory manner. Moreover, the application of cyclic tensile strain at low magnitudes results in a downregulated expression of MMPs under inflammatory conditions and an increased expression of TIMPs. This indicates that low force magnitudes could be a reason for bone and PDLs’ ECM formation at the tension site during OTM in the presence of inflammatory stimuli, which normally favor bone and PDL destruction [43,55]. To ensure the comparability of our findings with the two studies of Long et al. [43,55], we used the same low force magnitude (6%). In contrast to these studies, our data demonstrated opposite effects of mechanical forces on MMPs and TIMPs expression in hPDL-MSCs under inflammatory conditions, using tensile forces in a static mode. This indicates that the influence of tensile strains under inflammatory conditions on hPDL-MSCs also depends on its application form.

Our data showed that STS with 6% elongation alone had no influence on the hPDL-MSCs viability and on the MMP-1, *MMP-2* and *TIMP-1* expression. The influence of in vitro simulated orthodontic forces on the expression of multiple MMPs and TIMPs within the human PDL has been investigated in multiple studies, showing rather inconsistent results [5,8,25,28,30,56]. Our data are in accordance with the findings of Long et al. [43], who used the same force magnitude and comparable incubations times, but a cyclic tensile force application form. This indicates that applied orthodontic forces have no effects on the production of ECM remodeling enzymes within the tension zone of the human PDL in the absence of local aseptic inflammation. However, other studies demonstrated clear changes in the expression of MMP-1, MMP-2 and TIMP-1 in both directions when applying STS alone [8,25,30]. This inconsistency may result from the use of variable combinations of different parameters, such as force types (compression versus pressure), force magnitudes, application forms (static versus cyclic) and incubation times.

Since the effects of orthodontic forces on hPDL-MSCs are influenced by local aseptic inflammation, which is a prerequisite for OTM [2], we simulated this condition by adding different concentrations of IL-1β. Several clinical studies demonstrated that IL-β is one of the most abundant pro-inflammatory cytokines within the GCF during the initial OTM phase, at both the tension and the compression sites [38,39]. The chosen IL-1β concentrations and incubation times are in the range of previous studies, in which the authors used concentrations and time points ranging from 0.1 ng/mL to 10 ng/mL and from 4 to 48 h, respectively, to investigate the influence of IL-1β on the production of various MMPs [41,42,43]. 

In the absence of STS, our experiments demonstrated that both IL-1β concentrations significantly induce the production of MMP-1 and *MMP-2* in hPDL-MSCs in a time-dependent manner. This is in accordance with other studies, which also showed IL-1β induced expression of MMP-1, MMP-2 and MMP-3 [41,42,43]. These data, together with the fact that STS alone does not influence MMPs expression, suggest that the local aseptic inflammation is the major key for ECM remodeling within the PDL.

Our results also showed that *MMP-1* expression was markedly stronger induced by both IL-1β concentrations than *MMP-2* expression. On protein levels, MMP-2 was not detectable by ELISA. This finding is in accordance with Abe et al. [57], who conducted a study with gingival fibroblasts and observed that IL-1β induces a much higher expression of MMP-1 and that MMP-2 is also not detectable on protein level. 

IL-1β also regulates the breakdown of PDL collagens by affecting the expression of TIMPs, such as TIMP-1 and TIMP-2 [43], which are inhibitors of MMPs’ enzymatic activities [58]. Our experiments showed a stimulating effect of IL-β on the expression of *TIMP-1* by hPDL-MSCs in a concentration of 5 ng/mL, but not of 1 ng/mL. This corresponds to the data of Long et al. [43], who did not detect any effect of 1 ng/mL IL-β on *TIMP-1* expression. This indicated that higher local IL-1β levels have to be reached within the PDL to influence *TIMP-1* expression, whereas MMPs’ expression can be affected already by lower IL-1β concentrations. Hence, it can be suggested that the aseptic inflammation, especially the presence of IL-1β, possesses a dual role during OTM: low IL-1β levels at the beginning of inflammation favor PDL matrix degradation by increasing MMP-1 and MMP-2, whereas higher IL-1β levels at a later inflammation state additionally induces TIMP-1 expression and counteract MMPs-dependent matrix degradation.

More importantly, for the first time, we showed an enhancement of the IL-1β-induced MMP-1 and MMP-2 expression after applying STS for 24 h. It can be excluded that these increased MMPs levels result from an enhanced endogenous IL-1β production since high amounts of exogenous IL-1β were present during the experiment. Hence, it seems that tensile strain with 6% elongation, which was applied statically, favors ECM degradation within the PDL under inflammatory conditions. Our *TIMP-1* expression data support this conclusion, showing a negative effect of STS on IL-1β-triggered *TIMP-1* expression, which should favor MMPs’ activity. However, these data are different from Long et al. [43], showing an opposite effect of cyclic tensile strain with 6% elongation on IL-1β-induced MMPs and TIMPs production. Hence, it can be suggested that besides force magnitudes [43], the force application form could be a critical parameter that determines the effect of tensile strains on the production of MMPs and TIMPs in hPDL-MSCs and consequently on the ECM remodeling. However, it should be kept in mind that ECM remodeling is an interaction between numerous MMPs, TIMPs and other proteolytic enzymes [5,8,19], and not only the parameters investigated in this study.

In the second part of this study, for the first time, we directly compared the influence of 2% FBS on the STS-based effects on IL-1β induced expression of MMP-1, *MMP-2*, *TIMP-1* and *IL-1β*. This is especially important since FBS is known to affect hPDL-MSCs during in vitro cultivation [50]. Here we demonstrated that the hPDL-MSCs’ viability and the basal expression of investigated target genes are not affected by 2% FBS. In contrast, 2% FBS mainly caused a lower IL-1β-induced expression of *MMP-1, MMP-2* and *TIMP-1*, although IL-1β gene expression was clearly higher than in the absence of FBS. Most importantly, 2% FBS caused, on the one hand, lower and, on the other hand, opposite effects of STS on IL-1β-induced target gene expression. Hence, it is especially important to keep in mind the possible effects of FBS on MMPs and TIMPs synthesis in hPDL-MSCs when interpreting such results. Studies, which investigate MMPs and TIMPs expression in hPDL-MSCs, are very heterogeneous concerning the use of FBS while applying mechanical load on the cells. Numerous studies exist which use a broad range of FBS (1–10%) [24,28,42,46,55,56], whereas others do not use any FBS [5,25,41,43]. Additionally, multiple studies do not specify if FBS is added to cell culture during the experimental procedure [8,30,59,60]. This heterogeneity of FBS usage could be, besides using various combinations of force type, magnitude, application form and duration, another reason for the inconsistent results between several studies.

Extrapolating our results into the clinical situation suggests that not only the applied force magnitude but also the force application form (static versus cyclic) seems to be crucial for successful OTM. For example, nickel-titanium coil springs cause static forces during OTM [48,49]. Based on our results, it is possible that such static forces favor a pro-inflammatory environment at the tension site when applying low-magnitudes and, consequently, osteoclastogenesis [2,40]. Our results also indicate that such static forces may favor ECM degradation at the tension zone within the PDL. Hence, it is possible that nickel-titanium coil springs cause, at least at low-magnitude forces, a bone and PDLs’ ECM destruction, rather than the desired composition. On the contrary, multibracket appliance, together with occlusal forces, cause a cyclic orthodontic load [25,47], possibly having anti-inflammatory and osteogenic effects in the tension zone when using low-magnitude forces. Additionally, these cyclic tensile forces may inhibit ECM degradation in the tension area within the PDL [43]. Hence, when choosing materials for malocclusion treatment, orthodontists should be aware of their possible variable effects due to different force magnitudes and static and cyclic tensile forces caused by different materials [25,43,47,48,49]. Further, the choice of right orthodontic appliances may be especially important in adult patients, in which orthodontic treatment is a risk factor for periodontitis [61,62], since both have comparable molecular mechanisms and cause extensive PDL and alveolar bone remodeling [2,63,64]. However, it should be considered that periodontitis is characterized by a dysbiotic state and impaired host-microbe homeostasis [65,66,67].

## 4. Materials and Methods

### 4.1. Cell Culture

Third molars were extracted from periodontally healthy individuals due to orthodontic reasons and were used to isolate primary hPDL-MSCs. After the outgrowth of these cells from periodontal ligament slices, which were scrapped off from the mid-third of the tooths’ roots, hPDL-MSCs were cultivated in Dulbecco’s modified Eagle’s Medium (DMEM, Sigma-Aldrich, St. Louis, MO, USA), which was supplemented with 100 U/mL penicillin, 50 µg/mL streptomycin (both Sigma-Aldrich, St. Louis, MO, USA) and 10% FBS (Gibco, Carlsbad, CA, USA). hPDL-MSCs were cultured at 95% humidity, 5% carbon dioxide and at 37° Celsius. Passage levels five to seven were used for all conducted experiments. 

The mesenchymal stromal cell character of isolated primary hPDL-MSCs was verified according to the minimal criteria for mesenchymal stromal cells defined by the International Society for Cell and Gene Therapy (ISCT) [11]. The expression of hematopoietic and mesenchymal stromal cell surface markers was evaluated by immunostaining, as described in our previous studies [68,69].

The Ethics Committee of the Medical University of Vienna (EK Nr. 1079/2019, date of approval: 18 March 2019) approved the study protocol. All participating individuals got informed before the extraction procedure and gave their written consent. All experimental methods throughout the study were conducted according to the Declaration of Helsinki and the Good Scientific Practice Guidelines of the Medical University of Vienna.

### 4.2. Stimulation Protocol

50,000 hPDL-MSCs were seeded per well in 3 mL DMEM medium supplemented with 10% FBS, 100 U/mL penicillin and 50 µg/mL streptomycin at collagen type I coated 6-well BioFlex^®^ plates (FlexCell International Cooperation, Burlington, VT, USA). After 24 h incubation, cell cycles were synchronized by changing medium to DMEM without any FBS, supplemented with only 100 U/mL penicillin and 50 µg/mL streptomycin. After overnight-starvation, hPDL-MSCs were stimulated with 1 ng/mL or 5 ng/mL IL-1β (Invivogen, San Diego, CA, USA) using DMEM medium supplemented with either only antibiotics or with antibiotics and 2% FBS. Appropriate unstimulated hPDL-MSCs served as control. Simultaneously, STS was applied to hPDL-MSCs using the FlexCell FX-5000T™ Tension System (FlexCell International Cooperation, Burlington, VT, USA) with 6% equibiaxial elongation. IL-1β stimulated and unstimulated hPDL-MSCs, which were seeded on the same cell culture plate without applying STS served as references. 6 and 24 h later, MMP-1, MMP-2, TIMP-1 and IL-1β gene and protein expression levels were measured by quantitative polymerase chain reaction (qPCR) and ELISA, respectively. Additionally, cell viability was determined by live/dead staining followed by fluorescence microscopy 24 h after stimulation.

### 4.3. Cell Viability

Cell viability was determined 24 h after stimulation by live/dead staining using the Live-Dead Cell Staining Kit from Enzo (Enzo, Framingdale, NY, USA). After washing the adherent cells in cell culture plates with 1xPBS (phosphate buffered saline) three times, 1 mL staining solution (1 µL solution A + 1 µL solution B in 1 mL staining buffer) were added per well. After the incubation at 37 °C for 15 min, the fluorescence of green fluorescent Live-Dye™ (Ex/Em = 488/518 nm) and propidium iodide (Ex/Em = 488/515 nm) were determined by ECHO Revolve fluorescence microscope (Echo, San Diego, CA, USA) using four-fold magnification and an exposure time of 670 ms for both fluorescence dyes.

### 4.4. Quantitative Polymerase Chain Reaction

After cell harvesting, TaqMan Gene Expression Cells-to-CT kit (Applied Biosystems, Foster City, CA, USA) was used to prepare cell lysates, to reverse transcribe mRNA into cDNA and to perform qPCR. For reverse transcription, samples were heated to 37 °C for 1 h followed by 95 °C for 5 min using Primus 96 advanced thermocycler (PeqLab/VWR, Darmstadt, Hessen, Germany). qPCR was performed in paired reactions heating the samples to 1 × 95 °C for 10 min followed by 50 cycles of 15 s at 95 °C and 1 min at 60 °C. Target genes were amplified, and their expression levels were detected by a QuantStudio 3 device (Applied Biosystems, Foster City, CA, USA) using the following TaqMan Gene Expression Assays (all from Applied Biosystems, Foster City, CA, USA): *MMP-1*, Hs00899658_m1; *MMP-2*, Hs01548727_m1; *TIMP-1*, Hs99999139_m1; *IL-1β*, Hs01555410_m1 and *GAPDH*, Hs99999905_m1. After determining Ct values, the *n*-fold expression of *MMP-1*, *MMP-2* and *TIMP-1* compared to the appropriate controls was calculated using the 2^-∆∆*C*t^ method. The housekeeping gene *GAPDH* served as an endogenous reference. *IL-1β* gene expression levels are shown relative to *GADPH* expression without normalization to the appropriate control.

### 4.5. Immunoassay

MMP-1 protein levels were determined in conditioned media, which were harvested 6 and 24 h after stimulation. MMP-1 protein levels were evaluated by using Human MMP-1 ELISA Kit (Thermo Fisher Scientific, Waltham, MA, USA), following the manufacturer’s instructions. Provided standards ranged from 24.69 to 18,000 pg/mL. Absorbance was measured at OD_450nm_ and OD_570nm_ (optical density) followed by plotting OD_570_-subtracted values against the appropriate standard curve to calculate appropriate protein concentrations. MMP-2 and TIMP-1 protein production was not detectable in conditioned media.

### 4.6. Statistical Analysis

Data for each experiment were obtained from at least 4 independent repetitions with hPDL-MSCs isolated from at least 4 different individuals. All measured data are presented as mean values ± S.E.M. Statistical analysis was performed using SPSS (Version 24.0, IBM, Armonk, NY, USA). The normal distribution of the data was tested by the Kolmogorov-Smirnov test. For normally distributed data, statistical differences between groups were investigated by one-way analysis of variance (ANOVA) followed by post-hoc LSD test. In the absence of normal distribution, the analysis was performed using the Friedman Test, followed by the Wilcoxon test for pairwise comparison. *P*-values ≤ 0.05 were considered to be statistically significant.

## 5. Conclusions

In conclusion, this in vitro study demonstrated that applying STS with 6% elongation on hPDL-MSCs increased the expression of MMP-1, *MMP-2* and *IL-1β* under inflammatory conditions. Furthermore, *TIMP-1* expression decreased under this experimental setting. These data suggest that compared to a cyclic application, a static application form of low-magnitude forces may favor a more inflammatory and destructive local environment at the tension zone within the PDL. In further studies, these results have to be verified by directly comparing the static and cyclic application form of low-magnitude forces on hPDL-MSCs.

## Figures and Tables

**Figure 1 ijms-22-01027-f001:**
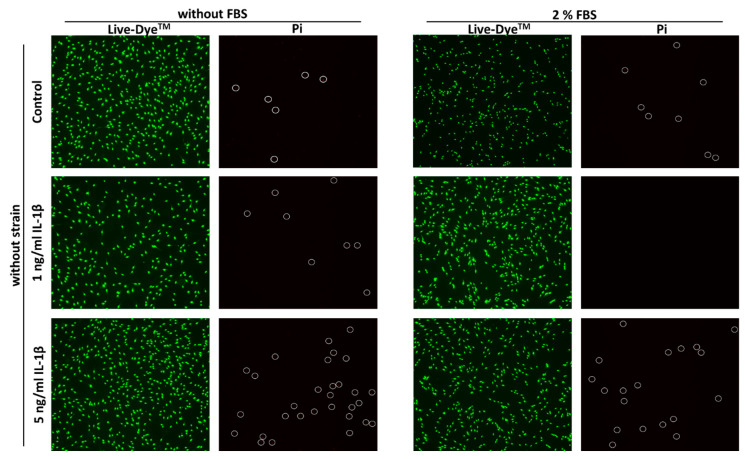
Representative pictures of live/dead staining of hPDL-MSCs after applying static orthodontic tensile forces under IL-1β-induced inflammatory conditions. After stimulating hPDL-MSCs with 1 ng/mL or 5 ng/mL IL-1β, STS with 6% equibiaxial elongation was applied in the absence or presence of 2% FBS. Unstimulated and non-stretched hPDL-MSCs served as references. 24 h after applying STS, hPDL-MSCs were doubled-stained with green fluorescent Live-Dye™ (Ex/Em = 488/518 nm) and propidium iodide (Pi) (Ex/Em = 488/515) to discriminate between living (green fluorescent) and dead (red fluorescent) hPDL-MSCs. Red fluorescent dead cells are indicated by white circles. Fluorescence was detected by ECHO Revolve fluorescence microscope using the FITC and Texas Red channels, a 4-fold magnification and an exposure time of 670 ms in both fluorescence channels.

**Figure 2 ijms-22-01027-f002:**
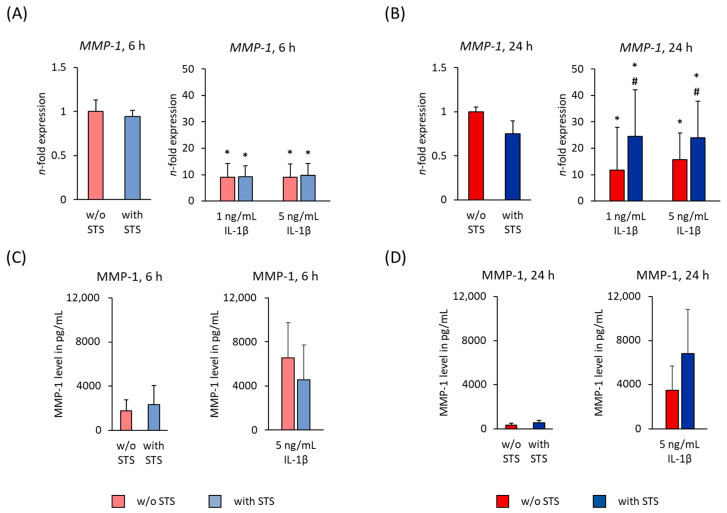
Effect of static orthodontic tensile forces on the IL-1β-induced gene and protein expression of MMP-1 in hPDL-MSCs in the absence of FBS. STS with 6% equibiaxial elongation was applied to untreated or IL-1β-treated hPDL-MSCs in the absence of FBS. 6 (**A**) and 24 (**B**) hours after stimulation with or without STS, qPCR was performed showing the *n*-fold expression levels of *MMP-1* compared to the appropriate controls (*n*-fold expression = 1). *GAPDH* served as an endogenous control. Corresponding MMP-1 protein levels were determined in pg/mL by enzyme-linked immunosorbent assay (ELISA) in conditioned media, which were harvested 6 (**C**) and 24 (**D**) hours after stimulation with or without STS. All data are presented as mean ± standard error of mean (S.E.M.). * *p*-value ≤ 0.05 significantly higher compared to the appropriate control; # *p*-value ≤ 0.05 significantly higher compared to the appropriate IL-1β stimulated cells without static STS.

**Figure 3 ijms-22-01027-f003:**
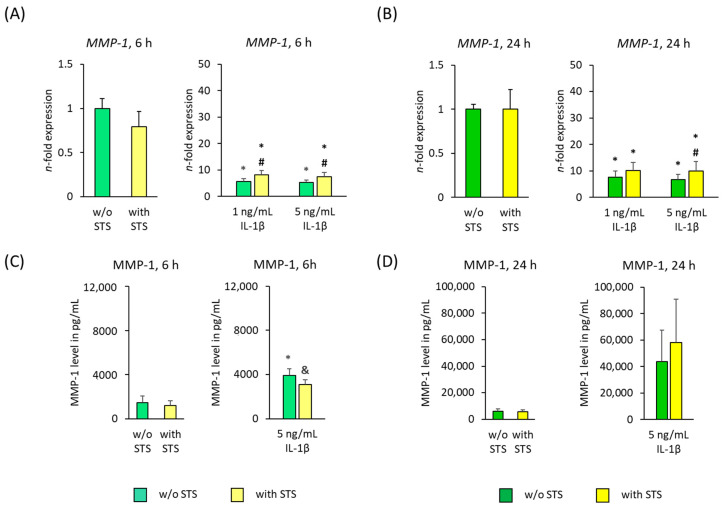
Effect of static orthodontic tensile forces on the IL-1β-induced gene and protein expression of MMP-1 in hPDL-MScs in the presence of 2% FBS. STS with 6% equibiaxial elongation was applied to untreated or IL-1β-treated hPDL-MSCs in the presence of 2% FBS. 6 (**A**) and 24 (**B**) hours after stimulation with and without STS, qPCR was performed showing the *n*-fold expression levels of *MMP-1* compared to the appropriate controls (*n*-fold expression = 1). *GAPDH* served as an endogenous control. Corresponding MMP-1 protein levels were determined in pg/mL by ELISA in conditioned media, which were harvested 6 (**C**) and 24 (**D**) hours after stimulation with or without STS. All data are presented as mean ± S.E.M. * *p*-value ≤ 0.05 significantly higher compared to the appropriate control; # *p*-value ≤ 0.05 significantly higher compared to the appropriate IL-1β stimulated cells without STS; & *p*-value ≤ 0.05 significantly lower compared to the appropriate Il-1β stimulated cells without STS.

**Figure 4 ijms-22-01027-f004:**
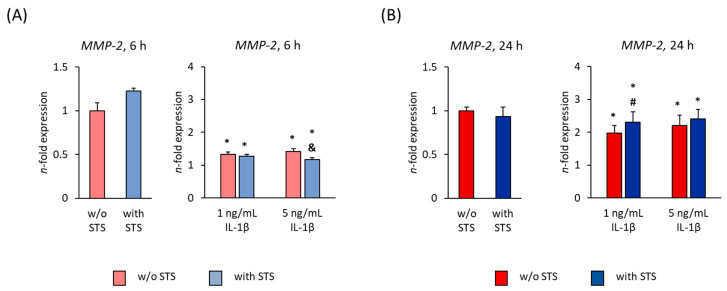
Effect of static orthodontic tensile forces on the IL-1β-induced gene expression of *MMP-2* in hPDL-MSCs in the absence of FBS. STS with 6% equibiaxial elongation was applied to untreated or IL-1β-treated hPDL-MSCs in the absence of FBS. 6 (**A**) and 24 (**B**) hours after stimulation with or without STS, qPCR was performed showing the *n*-fold expression levels of *MMP-2* compared to the appropriate controls (*n*-fold expression = 1). *GAPDH* served as an endogenous control. All data are presented as mean ± S.E.M. * *p*-value ≤ 0.05 significantly higher compared to the appropriate control; # *p*-value ≤ 0.05 significantly higher compared to the appropriate IL-1β stimulated cells without STS; & *p*-value ≤ 0.05 significantly lower compared to the appropriate Il-1β stimulated cells without STS.

**Figure 5 ijms-22-01027-f005:**
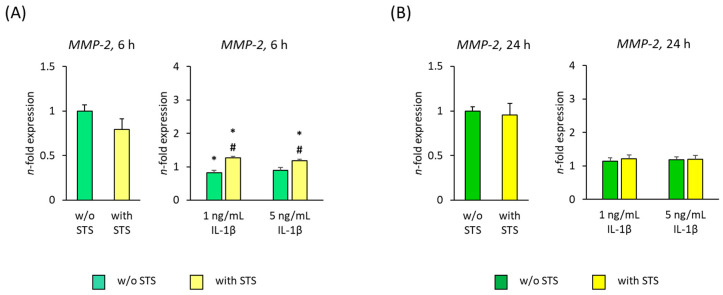
Effect of static orthodontic tensile forces on *MMP-2* gene expression in hPDL-MSCs in the presence of IL-1β and 2% FBS. STS with 6% equibiaxial elongation was applied to untreated or IL-1β-treated hPDLSCs in the presence of 2% FBS. 6 (**A**) and 24 (**B**) hours after stimulation with or without STS, qPCR was performed showing the *n*-fold expression levels of *MMP-2* compared to the appropriate controls (*n*-fold expression = 1). *GAPDH* served as an endogenous control. All data are presented as mean ± S.E.M. * *p*-value ≤ 0.05 significantly higher compared to the appropriate control; # *p*-value ≤ 0.05 significantly higher compared to the appropriate IL-1β stimulated cells without STS.

**Figure 6 ijms-22-01027-f006:**
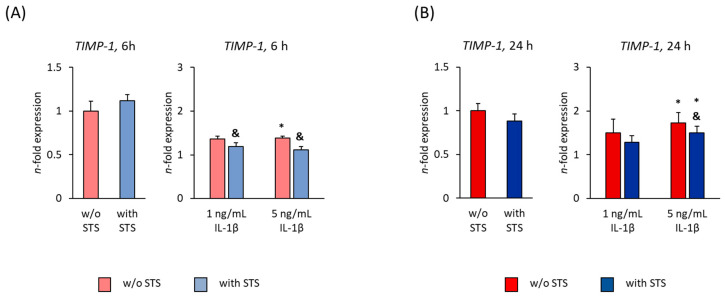
Effect of static orthodontic tensile forces on the IL-1β-induced gene expression of *TIMP-1* in hPDL-MSCs in the absence of FBS. STS with 6% equibiaxial elongation was applied to untreated or IL-1β-treated hPDL-MSCs in the absence of FBS. 6 (**A**) and 24 (**B**) hours after stimulation with or without STS, qPCR was performed showing the *n*-fold expression levels of *TIMP-1* compared to the appropriate controls (*n*-fold expression = 1). *GAPDH* served as an endogenous control. All data are presented as mean ± S.E.M. * *p*-value ≤ 0.05 significantly higher compared to the appropriate control; & *p*-value ≤ 0.05 significantly lower compared to the appropriate Il-1β stimulated cells without STS.

**Figure 7 ijms-22-01027-f007:**
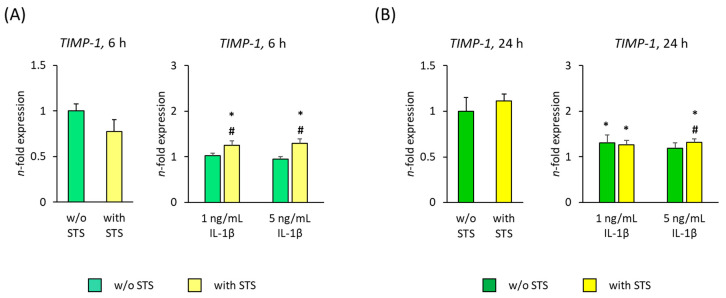
Effect of static orthodontic tensile forces on *TIMP-1* gene expression in hPDL-MSCs in the presence of IL-1β and 2% FBS. STS with 6% equibiaxial elongation was applied to untreated or IL-1β-treated hPDL-MSCs in the presence of 2% FBS. 6 (**A**) and 24 (**B**) hours after stimulation with or without STS, qPCR was performed showing the *n*-fold expression levels of *TIMP-1* compared to the appropriate controls (*n*-fold expression = 1). *GAPDH* served as an endogenous control. All data are presented as mean ± S.E.M. * *p*-value ≤ 0.05 significantly higher compared to the appropriate control; # *p*-value ≤ 0.05 significantly higher compared to the appropriate IL-1β stimulated cells without STS.

**Figure 8 ijms-22-01027-f008:**
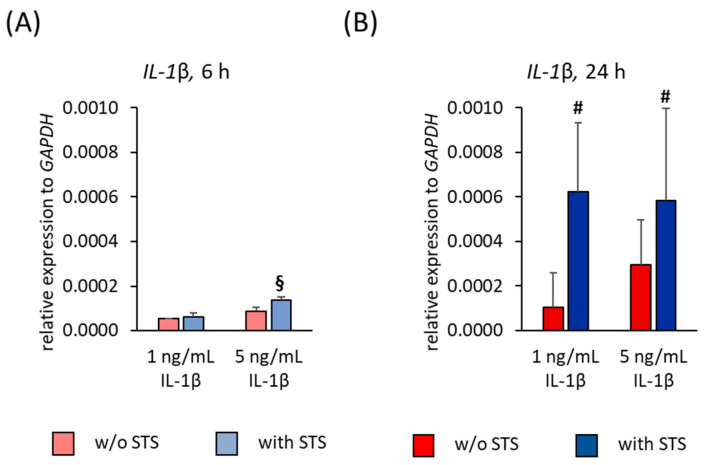
Effect of static orthodontic tensile forces on the IL-1β-induced gene expression of *IL-1β* in hPDL-MSCs in the absence of FBS. STS with 6% equibiaxial elongation was applied to untreated or IL-1β-treated hPDL-MSCs in the absence of FBS. 6 (**A**) and 24 (**B**) hours after stimulation with or without STS, qPCR was performed showing the relative *IL-1β* expression levels normalized to the internal reference gene *GAPDH*. All data are presented as mean ± S.E.M. # *p*-value ≤ 0.05 significantly higher compared to the appropriate IL-1β stimulated cells without STS; § *p*-value ≤ 0.05 significantly higher compared to 1 ng/mL IL-1β in the presence of STS.

**Figure 9 ijms-22-01027-f009:**
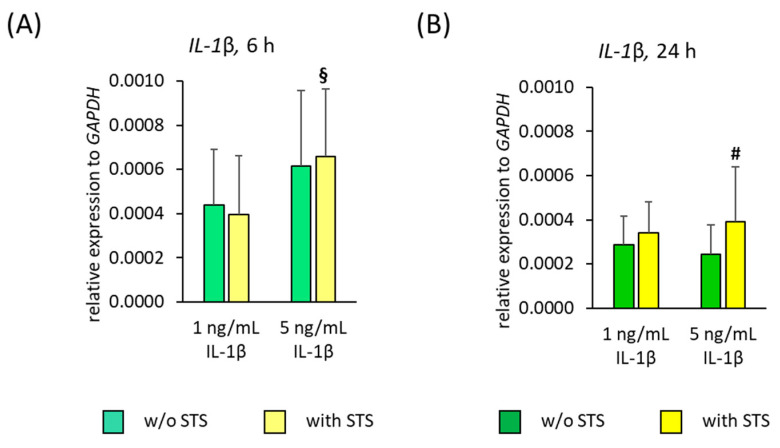
Effect of static orthodontic tensile forces on the IL-1β-induced gene expression of *IL-1β* in hPDL-MSCs in the presence of 2% FBS. STS with 6% equibiaxial elongation was applied to untreated or IL-1β-treated hPDL-MSCs in the presence of 2% FBS. 6 (**A**) and 24 (**B**) hours after stimulation with or without STS, qPCR was performed showing the relative *IL-1β* expression levels normalized to the internal reference gene *GAPDH*. All data are presented as mean ± S.E.M. # *p*-value ≤ 0.05 significantly higher compared to the appropriate IL-1β stimulated cells without STS; § *p*-value ≤ 0.05 significantly higher compared to 1 ng/mL IL-1β in the presence of STS.

## Data Availability

The data presented in this study are available on request from the corresponding author.

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
