# Peer review of "Interleukin-1β Induced Matrix Metalloproteinase Expression in Human Periodontal Ligament-Derived Mesenchymal Stromal Cells under In Vitro Simulated Static Orthodontic Forces"

_ijms, 2021, doi:10.3390/ijms22031027_

Round 1

Reviewer 1 Report

In the manuscript entitled: “Interleukin-1β induced matrix metalloproteinase expression in human periodontal ligament derived mesenchymal stromal cells under in vitro simulated static orthodontic forces” the authors investigated how IL-1β induced expression of MMPs, TIMPs and IL-1β in hPDL-MSCs after mimic static orthodontic tensile forces with low magnitude

The author's found that, after 6- and 24 hours, MMP-1, MMP-2, TIMP-1 and IL-1β expression levels were measured. In the absence of FBS, STS alone had no influence on the basal expression of investigated target genes, whereas IL-1β caused increased expression of these genes. In combination, they increased the gene and protein expression of MMP-1 and the gene expression of MMP-2 after 24 hours. After 6 hours, STS reduced IL-1β-induced MMP-1 synthesis and MMP-2 gene expression. IL-1β-induced TIMP-1 gene expression was decreased by STS after 6- and 24-hours. At both time points, the IL-1β-44 induced gene expression of IL-1β was increased.

The authors concluded that applying STS with 6% elongation on hPDL- MSCs increased and decreased the expression of MMP-1/2 and TIMP-1, respectively, under IL-1β- induced inflammatory conditions.

Major comments:

In general, the idea and innovation of this study, regards the analysis of periodontal ligament and their mediators is interesting, because the role of mediators released during periodontal disease are validated, but further studies on this topic could be an innovative issue in this field could be open a creative matter of debate in literature by adding new information. Moreover, there are few reports in the literature that studied this exciting topic with this kind of study design.

The study was well conducted by the authors; However, there are some concerns to revise that are described below.

The introduction section and the manuscript resume the existing knowledge regarding the important factor linked with mediators released during inflammation.

However, as the importance of the topic, the reviewer strongly recommends, before a further re-evaluation of the manuscript, to update the literature through read, by must discuss and cites in the references with great attention all of those recent interesting articles, that helps the authors to better introduce and discuss the aim of the study in light of others mediators released during periodontitis: 1) Isola G, Lo Giudice A, Polizzi A, Alibrandi A, Murabito P, Indelicato F. Identification of the different salivary Interleukin-6 profiles in patients with periodontitis: A cross-sectional study. Arch Oral Biol. 2020 Nov 30;122:104997. doi: 10.1016/j.archoralbio.2020.104997. 2) Isola G, Polizzi A, Patini R, Ferlito S, Alibrandi A, Palazzo G. Association among serum and salivary A. actinomycetemcomitans specific immunoglobulin antibodies and periodontitis. BMC Oral Health. 2020 Oct 15;20(1):283. doi: 10.1186/s12903-020-01258-5. 3) Isola G, Polizzi A, Alibrandi A, Williams RC, Leonardi R. Independent impact of periodontitis and cardiovascular disease on elevated soluble urokinase-type plasminogen activator receptor (suPAR) levels. J Periodontol. 2020 Oct 22. doi: 10.1002/JPER.20-0242.

The authors should be better specified at the end of the introduction section, the rationale of the study. In the discussion, should better clarify the cell culture and the stimulation protocol performed.

The conclusion should be added with the main findings of the study and reinforce in light of the future directions.

In conclusion, I am sure that the authors are excellent clinicians who achieve very nice results with their adopted protocol. However, this study, in my view, does not in its current form satisfy a very high scientific requirement for publication in this journal and requests a revision before a further re-evaluation of the manuscript.

Minor Comments:

Introduction:

  • Please refer to major comments

Discussion

  • Please add a specific sentence that clarifies the results obtained in the first part of the discussion

Author Response

Reviewer 1

Comment 1

In general, the idea and innovation of this study, regards the analysis of periodontal ligament and their mediators is interesting, because the role of mediators released during periodontal disease are validated, but further studies on this topic could be an innovative issue in this field could be open a creative matter of debate in literature by adding new information. Moreover, there are few reports in the literature that studied this exciting topic with this kind of study design. The study was well conducted by the authors; However, there are some concerns to revise that are described below.

Authors’ answer

We thank reviewer 1 for this positive feedback concerning the idea of this manuscript and we appreciate all of her/his comments.

Comment 2

The introduction section and the manuscript resume the existing knowledge regarding the important factor linked with mediators released during inflammation. However, as the importance of the topic, the reviewer strongly recommends, before a further re-evaluation of the manuscript, to update the literature through read, by must discuss and cites in the references with great attention all of those recent interesting articles, that helps the authors to better introduce and discuss the aim of the study in light of others mediators released during periodontitis: 1) Isola G, Lo Giudice A, Polizzi A, Alibrandi A, Murabito P, Indelicato F. Identification of the different salivary Interleukin-6 profiles in patients with periodontitis: A cross-sectional study. Arch Oral Biol. 2020 Nov 30;122:104997. doi: 10.1016/j.archoralbio.2020.104997. 2) Isola G, Polizzi A, Patini R, Ferlito S, Alibrandi A, Palazzo G. Association among serum and salivary A. actinomycetemcomitans specific immunoglobulin antibodies and periodontitis. BMC Oral Health. 2020 Oct 15;20(1):283. doi: 10.1186/s12903-020-01258-5. 3) Isola G, Polizzi A, Alibrandi A, Williams RC, Leonardi R. Independent impact of periodontitis and cardiovascular disease on elevated soluble urokinase-type plasminogen activator receptor (suPAR) levels. J Periodontol. 2020 Oct 22. doi: 10.1002/JPER.20-0242.

Authors’ answer

We have added information about periodontitis and involved biological mediators, such as interleukin-6 in the introduction section (S. 2, lines 93-94). Further, in the discussion section we shortly described the connection between orthodontic treatment and periodontitis. We shortly mention commonalities and differences in the underlying cellular mechanisms (S. 13, lines 453-457). Within these two added sections we added two of the requested references (Isola G et al. 2020 Nov. Arch Oral Biol. and Isola G et al. 2020 Oct. BMC Oral Health). We did not include the third requested paper (Isola G et al. 2020 Oct. J Periodontol.) in our revised manuscript since this reference is not entirely within the scope of our manuscript.

Comment 3

 The authors should be better specified at the end of the introduction section, the rationale of the study.

Authors’ answer

We thank the reviewer for this comment. We carefully revised the appropriate paragraph in the introduction section (S. 2-3, lines 88-101) and tried to better highlight the rationale of this study. Although studies already exist which investigate MMPs and TIMPs expression in PDL cells under inflammatory conditions, these studies used low-magnitude forces in only a cyclic application form. However, this application form mimics only certain orthodontic appliances (such as multibracket appliances). Other appliances, such as nickel-titanium coil springs, cause static orthodontic forces, which have to be simulated in vitro in a static application form. We have added this information as rationale for this study (S. 3, lines 99-101). Furthermore, the use of a static application form in our experimental design is explained in more detail in the first paragraph of the discussion section (S. 10-11, lines 346-353).

Comment 3

In the discussion, should better clarify the cell culture and the stimulation protocol performed.

Authors’ answer

The rationale why we used certain parameters in our experimental design has been mostly enclosed within the discussion section. We have highlighted the appropriate information in the revised version of the manuscript. Additionally, we have carefully revised appropriate paragraphs to better highlight the information and also added some new information:

Static application mode (S. 10-11, lines 346-353): we have revised the paragraph to better highlight the rationale to use this force application mode.

6% elongation (S. 11, lines 363-366): We used 6% elongation to ensure comparability of our study with two previous studies (Long et al. 2001 and Long et al. 2002), which used the same percentage of elongation to apply tensile strain, however, in a cyclic force application mode. We have added this information to the appropriate paragraph in the discussion section.

5 ng/ml and 1 ng/ml IL-1β and incubation times (S. 11, lines 381-388): The information why we used IL-1β has been already included in the discussion section of the submitted manuscript. We have highlighted the appropriate paragraph in the revised manuscript version. Several studies (Xiang et al. 2009, Murayama et al. 2011 and Long et al. 2002), which investigated the influence of IL-1β on the production of various MMPs have used concentrations ranging from 0.1 ng/ml to 10 ng/ml and incubation times ranging from 4 to 48 hours. Based on these previous studies, we decided to use 5 and 1 ng/ml IL-1β and 6 to 24 hours incubation times, which both are in the range of these previous studies. We have added this information to the appropriate paragraph in the discussion section.

Comment 4

The conclusion should be added with the main findings of the study and reinforce in light of the future directions.

Authors’ answer

We have transferred the conclusion paragraph of the discussion section to the end of the manuscript into an extra conclusion section (S. 14, lines 530-536). Further we have revised the transferred paragraph to highlight the main findings of our study. Additionally, we added information about the future directions (S. 14, lines 535-536).

Comment 5

Please add a specific sentence that clarifies the results obtained in the first part of the discussion.

Authors’ answer

We have specified the main findings of the first part of our study in the first paragraph of the discussion section (S. 11, lines 353-355).

Reviewer 2 Report

In this work the authors investigate the primary role of hPDL-MSC when were stimulated with IL-1β in combination with 38 static tensile strains (STS) with 6% elongation in the absence and presence of 2% fetal bovine serum 39 (FBS).

It is strongly suggest to revise the abstract in order to be clear.

The authors should add in the abstract section the aim of the study.

In the results section:

cell viability results, please add an histogram with the percentage of live and dead cells. 

Figure 2 C and D : the authors declared any significance. is that real? it doesn't look like this. please double check.

Figure 3 D : the authors declared any significance. is that real? please double check.

The authors should add a panel with the characterization of hPDL-MSC.

in the discussion section, the Authors should rearrange the discussion section in a more clarity way to better focus their topic.

the conclusion section is missing, please add.

The manuscript needs some revision by a native English Speaker.

Author Response

Reviewer 2

Comment 1

It is strongly suggest to revise the abstract in order to be clear. The authors should add in the abstract section the aim of the study.

Authors’ answer

We carefully revised the abstract to make it more intelligible. The introduction section of the abstract was especially revised. Additionally, we added a sentence where we explained the aim of this study (S. 1, lines 35-37).

Comment 2

cell viability results, please add an histogram with the percentage of live and dead cells.

Authors’ answer

To distinguish between living and dead hPDL-MSCs we used the Live-Dead Cell Staining Kit from Enzo followed by fluorescence microscopy as a qualitative approach. The aim of this experiment was to show that our experimental conditions have no apoptotic effect on hPDL-MSCs. We could prove it, because as can be seen, most of the cells are viable at the end of the experiments at any condition. Unfortunately, living and dead cells cannot be counted with our microscope-based software to get a quantitative output, too.

Comment 3

Figure 2 C and D : the authors declared any significance. is that real? it doesn't look like this. please double check.       
Figure 3 D : the authors declared any significance. is that real? please double check.

Authors’ answer

We thank the reviewer for addressing this important point. We have double-checked the appropriate statistical analysis. We found no mistakes. No statically significant differences were observed. The appropriate p-values were near the border of 0.05, namely: Figure 2C: p-value 0.171; Figure 2D: p-value 0.248; Figure 3D: 0.198. We have added these p-values to the appropriate result paragraphs (S. 4-5, lines 147-150 and S.5, line 174).

Comment 4

The authors should add a panel with the characterization of hPDL-MSC.

Authors’ answer

We created a table which shows the results of the flow cytometry analysis of MSCs’ and hematopoietic surface marker expression in hPDL-MSCs. The table shows the percentage of positive hPDL-MSCs for each listed surface marker. The data are presented as mean ± standard error of the mean. We included this table in the supplementary information as Supplementary Table S1. Additionally, we added a new paragraph in the results section (S.3 lines 110-115) describing supplementary table S1.

Comment 5

in the discussion section, the Authors should rearrange the discussion section in a more clarity way to better focus their topic.

Authors’ answer

We have revised the discussion section, rearranged it and shortened it by about 20%. We hope that the revised discussion is more transparent and comprehensible.

Comment 6

the conclusion section is missing, please add.

Authors’ answer

In our submitted manuscript we included the conclusion paragraph into the discussion section. In the revised manuscript we transferred this paragraph into an extra conclusion section at the end of the manuscript (S. 14, lines 530-536). Additionally, we revised the conclusion paragraph in accordance to the comments of reviewer 1 (see comment 4, reviewer 1).

Comment 7

The manuscript needs some revision by a native English Speaker.

Authors’ answer

The manuscript was checked by a native English Speaker and the whole manuscript was revised accordingly to the instructions of the native English Speaker.

Round 2

Reviewer 1 Report

The authors have well addressed to all reviewer's comments.

Reviewer 2 Report

The authors of the manuscript have been revised accordingly the ms,
addressing point by point to all the comments.

The english has been revised by a native speaker,
now the manuscript results more clearer and understandable.

please in the conclusion section correct the word "magntiude" line 541

please in the conclusion section correct the word "Furhtermore" line 539

minor spelling mistakes and punctuation errors need to be revised